# A systematic review assessing the quality of patient reported outcomes measures in dry eye diseases

Alberto Recchioni[1,2,3☯], Olalekan Lee Aiyegbusi[1,4☯], Samantha Cruz-Rivera[1☯], Saaeha Rauz[2☯]*, Anita Slade[1,4,5☯]

1 Centre for Patient Reported Outcomes Research, Institute of Applied Health Research, University of Birmingham, Birmingham, United Kingdom, 2 Academic Unit of Ophthalmology, Institute of Inflammation and Ageing, Birmingham and Midland Eye Centre, University of Birmingham, Birmingham, United Kingdom, 3 Optometry & Vision Sciences Group, School of Life & Health Sciences, Aston University, Birmingham, United Kingdom, 4 National Institute for Health Research (NIHR) Birmingham Biomedical Research Centre, University Hospitals Birmingham NHS Foundation Trust and the University of Birmingham, Birmingham, United Kingdom, 5 National Institute for Health Research Surgical Reconstruction and Microbiology Research Centre, West Midlands, UK University, Birmingham, United Kingdom

☯ These authors contributed equally to this work.
* S.RAUZ@bham.ac.uk

**Data Availability Statement:** All relevant data are within the paper and its Supporting Information files.

## Abstract

### Background

Patient-reported outcome measures (PROMs) can provide valuable insights on the impact of a disease or treatment on a patient's health-related quality of life. In ophthalmology, particularly in dry eye disease (DED) and ocular surface disease (OSD), it is unclear whether the available PROMs were developed using comprehensive guidelines. To address this, we evaluated the methodological quality of studies assessing the psychometric properties of PROMs in DED and OSD [PROSPERO registration number CRD42019142328].

### Methods

Four databases were searched; reference list and citation searching of included studies was also conducted. The COnsensus-based Standards for the selection of health Measurement INstruments (COSMIN) checklist was used to appraise the quality of the studies evaluating the psychometric properties of PROMs used in DED and OSD.

### Results

The search strategy (S3 Table) retrieved 5,761 records, 573 duplicates were removed, 5,188 abstracts were screened and 127 full-text articles were retrieved for further review. Of these, 118 full-text articles did not meet the eligibility criteria and were excluded. Reference list and citation searching, identified an additional 8 articles bringing the total numbers of papers reviewed to 17. In general, psychometric properties such as content validity, measurement error and structural validity were not assessed by the studies included in this review. Studies reviewing The Impact of Dry Eye on Everyday Life (IDEEL) presented with

**Funding:** This work was supported by funding from the II-LA-1117-20001 Programme Invention for Innovation (i4i) and it presents an independent research supported by the National Institute of Health Research Birmingham Biomedical Research Centre at the University Hospitals Birmingham National Health Service Foundation Trust and the University of Birmingham. The study funders did not have any role in the study design; the collection, analysis, and interpretation of the data; the writing of the report; or the decision to submit the article for publication. AR is supported by funding from the II-LA-1117-20001 Programme Invention for Innovation (i4i), National Institute for Health Research (NIHR). OLA is supported by the National Institute of Health Research (NIHR) Birmingham Biomedical Research Centre (BRC), West Midlands, Birmingham and reports funding by the Health Foundation and UCB Biopharma. He reports personal fees from Gilead Sciences Ltd. SCR is supported by the funding from Consejo Nacional de Ciencia y Tecnologia (CONACYT) (the National Council of Science and Technology CONACYT) of Mexico. SR is supported by National Institute for Health Research (NIHR) Programme Invention for Innovation (i4i), Medical Research Council (MRC) Developmental Pathway Funding Scheme (DPFS), Fight for Sight, Sandwell and West Birmingham Hospitals National Health Service Trust. AS is supported by funding from the II-LA-1117-20001 Programme Invention for Innovation (i4i) and reports funding by: British Heart Foundation, National Institute for Health Research Invention for Innovation, Medical Research Council, National Institute for Health Research MedTech and INvitro diagnostic Co-operative – Trauma management, Eye Hope Foundation, Association du Syndrome de Wolfram, Snow Foundation; Pfizer Health Research Foundation; National Institute for Health Research Clinical Research Network; National Institute For Health Research, Birmingham Biomedical Research Centre and National Institute For Health Research Surgical Reconstruction and Microbiology Research Centre at the University of Birmingham and University Hospitals Birmingham National Health Service Foundation Trust outside the submitted work.

**Competing interests:** OLA reports personal fees from Gilead outside the given work. This does not alter our adherence to PLOS ONE policies on sharing data and materials.

the highest quality scores together with the Ocular Surface Disease Index (OSDI) questionnaire.

## Conclusions

The quality of studies evaluating PROMs in DED and OSD was considered using the COSMIN standards. The majority of the studies evaluating PROMs included in this review did not meet the recommended COSMIN criteria and the quality of the PROMs evaluated is not assured. Further evaluation of their psychometric properties is required if these are going to be used in clinical practice or research.

## Introduction

Dry eye disease (DED) is a multifactorial disease of the anterior segment of the eye in which the tear film homeostasis fails, leading to ocular symptoms, and signs such as ocular staining, tear film instability and inflammation [1]. The most common cause of DED is inflammation caused by a group of conditions known as ocular surface disease (OSD) [2] that include auto-immune driven diseases such as Sjögren's Syndrome or Ocular Mucous Membrane Pemphigoid. This can result in a range of eye symptoms including dryness, irritation, poor vision, sore or fatigued eyes, and watering. Because of the nature of the symptoms, quality of life can be affected and the individual may be unable to carry out activities of daily living including work, socialisation, and driving [3], or maybe unable to sleep due to excessive dryness. DED has a global prevalence ranging from 20% to 50% worldwide (depending on whether signs and symptoms are studied separately or together), and has also been reported as one of the most common reasons for eye consultations [4]. Gender (female), older age groups, ethnicity, computer use and some medications (antihypertensives, antihistamines, and antidepressants) are commonly reported risk factors [4].

There is a wide selection of treatments available for DED including lubricant eye drops, ointments, gels, liposomal sprays, commercially available lid warming devices, anti-inflammatory eye drops, therapeutic contact lenses, punctal plugs, and serum eye drops [5,6]. Lubricants can range from simple formulations with the aim of increasing the tear film volume and reducing friction (e.g. hypromellose drops) to those which claim to restore parts of the tear film structure (e.g. liposome drops). Agents with differing viscosities variably purport to act as protective shields in front of the ocular surface (gels) or oil content that can be used when asleep (e.g. ointments). Biological tear substitutes are unlicensed medicines prepared from the patient's own blood (autologous serum eye drops) or from healthy donors (allogeneic serum eye drops). These are reserved for patients with the severest form of DED and provide nutritive benefits that replicate in part, the chemical constituents of the tear film which the commercially available lubricants to address. Many patients symptomatically benefit from serum eye drop use but access is limited due to regulations and funding restrictions. The nature of the condition requires frequent administration of the topical treatments and this can itself bring issues on patient well-being. The storage requirements such as keeping blood products frozen (e.g. autologous and allogeneic serum drops), can be challenging particularly those who wish to travel large distances using public transportation [7]. Therapeutic options are not curative; DED is a chronic condition that can negatively impact on a patient's quality of life because of chronic pain, depression, and secondary sleep disorders [8]. Time to trade off analyses have ranked living with DED similar to moderate angina or patients on renal dialysis [9].

In the last few years, patient reported outcome measures (PROMs) have been increasingly used in research and clinical trials to capture the efficacy of an intervention or a treatment [10]. The use of PROMs in clinical trials has also been shown to identify additional benefits or harms that might have been missed using clinical data alone [11]. PROMs are also important for putting patients at the heart of clinical research and routine clinical practice [12]. They can be used to track the condition from a patient's perspective and to understand how their health-related quality of life (HRQoL) is being affected by their health and therapeutic interventions [13]. Studies have also shown that PROMs that have been developed with input from people living with the condition, are more robust and sensitive to change over time [14].

In ophthalmology, there are several PROMs focused on the impact of visual impairment due to a condition (e.g. glaucoma, retinopathy, cataract), or tracking the outcome of refractive surgery (cornea or lens surgery) and low vision rehabilitation [15]. However, there is uncertainty about the psychometric quality of the PROMs available. In fact, in the field of dry eye and ocular surface disease, it is unclear whether the available PROMs were developed following guidelines such as those published by the Food and Drug Administration (FDA) [16] (S2 Table) and the European Medicines Agency (EMEA). This poses issues when using PROMs to evaluate outcomes of novel devices or therapeutics where patient perceptions of their disease and benefits of treatment are a critical outcome for the success of the technology being adopted into clinical practice. Identifying the best PROMs to be used in clinical practice or research, we need to evaluate the psychometric properties of PROMs available for use. Grubbs et al. [17] reviewed some of the most common questionnaires in assessing the quality of life (QoL) in DED in 2014. They concluded that the Ocular Surface Disease Index (OSDI) and the Impact of Dry Eye in Everyday Life (IDEEL) were the most reliable measures. The term "ocular surface disease" was not included in the search strategy. "Ocular surface disease" is defined as a group of disorders, of diverse pathogenesis, in which disease results from the failure of mechanisms responsible for maintaining a healthy ocular surface. The term indicates damage to the surface layers of the eye and includes conditions with systemic comorbidities such as rheumatoid arthritis, Sjögren's syndrome, mucous membrane pemphigoid, Steven-Johnson syndrome, atopic keratoconjunctivitis that can lead to ocular surface failure and blindness. Other conditions include infective keratitis, and ocular surface tumours. While 'Dry eye disease' is a feature of some but not all forms of ocular surface disease, systemic comorbidities amongst those who have dry eye disease, can impact upon the patients' tolerance of DED symptoms. The difficulty in delivering arduous treatment regimens in these patients impact upon patient wellbeing e.g. limited hand function can impinge upon their ability to self-administer eye drops and increases dependence on others. This may not be generalisable to other forms of ocular surface disease.

The Grubbs et al. [17] review did not use a validated tool such as the Consensus-based Standards for the selection of health Measurement INstruments (COSMIN) checklist to benchmark their evaluation of the included studies. Mokkink et al. [18] described COSMIN guidelines as a useful tool for the evaluation of the methodological quality of studies assessing the psychometric properties of PROMs. The guidelines were developed between 2006 and 2007 with input from a Delphi panel of more than 50 international experts who identified the key items and measurement properties captured by the guidelines. The guidelines were also updated in 2018 [19] when the authors decided to align with the Cochrane Collaboration for systematic reviews of intervention studies [20] by including inadequate studies which were previously excluded to avoid potential bias in their results. Additionally, the updated version removed reference to reasonable gold standard regarding criterion validity and responsiveness. The COSMIN panel reached a consensus that currently no gold standard exists for PROMs. Finally, the updated version removed references to sample size standards except for

adequate sample size for Content Validity, Structural Validity and Cross-cultural validity. Using a comprehensive review such as the COSMIN guidelines can help determine which psychometric properties were considered important and the best way to evaluate these using a standardised approach. These guidelines are not only useful for research settings but clinicians can also benefit by using this practical tool to evaluate the potential of outcome measurement instruments in practice. Using these guidelines to evaluate the quality of the studies ensures a consistent approach for evaluating the quality of studies evaluating PROMs and allows structured comparisons to be carried out.

Therefore, we present a systematic review of the quality of studies that report an evaluation of the psychometric properties of PROMs developed for use in patients with dry eyes and ocular surface diseases following the latest COSMIN guidelines [18] (S1 Table).

## Methods

### Design

This systematic review was registered with PROSPERO (registration number CRD42019142328). It was reported following the Preferred Reporting Items for Systematic Review and Meta-Analysis (PRISMA) checklist [21].

**Search strategy.** The following electronic databases were searched from inception: MEDLINE (Ovid) and EMBASE (Ovid), PsycINFO (Ovid) and CINAHL Plus (EBSCO). All the records were uploaded to Endnote X9 (Thomson Reuters). There were no restrictions in terms of publication period or language.

The search terms included in the search strategy were agreed with two different clinicians in the field of dry eyes and ocular surface disease. The sensitivity search filter developed by Terwee et al. [22] was adapted and used on all the databases (S3 Table). In addition, papers included in the full-text screening process were subjected to a hand search of reference lists which has been conducted using Web of Science (WoS). The systematic review searches have been done in 31.01.2020 and rechecked on 07.12.2020.

**Selection of studies.** Studies were included if they focused on PROMs used specifically in patients with dry eyes and ocular surface disease. Additionally, the following eligibility criteria were considered:

Inclusion criteria

1. Articles reporting PROM development in all dry eye and ocular surface disease populations.

2. Articles reporting the assessment of one or more psychometric properties for PROM(s) in all dry eye and ocular surface disease populations.

    Exclusion criteria

1. Studies reporting the psychometric validation of clinician-reported instruments.

2. Trials or studies evaluating the effectiveness of a treatment or intervention where a PROM questionnaire is used as an endpoint.

3. Editorials, reviews and conference abstracts.

All titles and abstracts were screened by 3 independent reviewers (AR, OLA and SCR). If an abstract did not provide enough information to allow the reviewers to make a judgment and take a decision, the abstract was included in the full-text screening process to avoid the risk of missing potentially eligible articles. Full-text articles were retrieved for studies that met the eligibility criteria and then reviewed by the same investigators.

Reasons for exclusions at the full-text screening process were recorded. At any stage, if the reviewers were unable to reach a consensus, a fourth reviewer was consulted (AS or SR).

A PRISMA flowchart summarises the process of selection (S1 Fig).

**Data extraction.** Two independent reviewers (AR and OLA) independently extracted the data from each study using a data collection form, with disagreements resolved through discussion and, if necessary, consultation with one of the other reviewers (AS and SR).

**Assessment of methodological quality.** COSMIN guidelines were used in this review to evaluate the quality of the studies evaluating the psychometric properties of PROMs used in DED and OSD. Following the data extraction, the methodological quality of the papers considered in the systematic review was assessed by two reviewers (AR and OLA) using the latest version of the validated COSMIN checklist [23]. The structure of the COSMIN checklist is divided into 10 properties for evaluation: PROM development, content validity, structural validity, internal consistency, cross-cultural validity, reliability, measurement error, criterion validity, hypotheses testing for construct validity and responsiveness. Every property can be rated individually using the COSMIN 4-point scale as "very good", "adequate", "doubtful" and "inadequate". Additionally, if a study has not considered one of the quality items rated with the COSMIN checklist, the answer "data not available" has been recorded. The overall score of the quality is determined by taking the lowest rating of any box considered (e.g. "the worst score counts" principle) [24].

In case of discrepancies between the reviewers (AR and OLA), the overall score was discussed with another reviewer (AS and SR).

## Results

This section presents information on the measurement properties that were reported by the individual studies including properties such as content validity, cross cultural validity, measurement invariance, measurement error and criterion validity.

A total of 5,761 references were identified, duplicates were removed (n = 573) and 5,061 references were excluded following the inclusion and exclusion criteria. Therefore, 127 full-text records were assessed for eligibility, of which only 17 were reviewed. The selection process is detailed in the PRISMA flow diagram (S1 Fig). A summary of the characteristics of the included studies and PROMs were summarised in Table 1.

Table 2 summarises the characteristics of the included studies.

All the studies considered were single studies and measurement properties are summarised in Table 3.

Table 4 includes the Updated COSMIN criteria for good measurement properties.

None of these PROMs were previously appraised using COSMIN guidelines according to the COSMIN database and other available databases. The included studies have evaluated a range of different properties such as development and validation, construct validity, psychometric evaluation, minimal clinically importance differences, and reliability, etc.

In total, 3,350 subjects with or without DED diagnosis were recruited to the included studies (sample sizes ranged from 33 to 907). All participants were adults with ages ranging from 18 to 86 years. Studies were heterogeneous in terms of population demographic characteristics (age, gender and ethnicity); however, the number of female participants included in the studies was larger than the male participants (16 out of 17 studies included) this is consistent with the aetiology of the condition. The Ocular Surface Disease Index (OSDI) (n = 3) and the Impact of Dry Eye on Everyday Life (IDEEL) (n = 3), were the most frequently evaluated measures followed by the Symptom Assessment in Dry Eye (SANDE) (n = 2), and the University of North Carolina Dry Eye Management Scale (UNC DEMS) (n = 2). S1 Table provides information about the COSMIN definition and criteria used in this study.

**Table 1. Characteristics of included studies.**

| | Study | Type of study | Study population and setting | Country/ Language | Sample size (% females) | Mean age in years (SD or range) | Duration of condition/ treatment Months, Mean (SD) unless otherwise stated |
|---|---|---|---|---|---|---|---|
| 1 | Miller et al. 2010 [25] | Examine psychometric properties (OSDI responsiveness) | Patients with dry eyes | USA/English | 310 (81.6) | 57.8 | Not specified |
| 2. | Schiffman et al. 2000 [26] | Psychometric evaluation study of dry eye symptoms | Patients with dry eyes | USA/English | 139 (total) 30 (control) (83.3) 109 (Dry eye) (78.9) | 57.5 (13.9) (control) 54.5 (13.4) (Dry eye) | Dry eye for at least 3 months |
| 3. | Dougherty et al. 2011 [27] | Psychometric evaluation study of dry eye symptoms (OSDI structural validity) | Patients with dry eye | USA/English | 172 (100) | 63 (8) | Not specified |
| 4. | Abetz et al. 2011 [28] | Development and validation study of dry eye symptoms and aspects of patients' daily life | Non-Sjögren's Syndrome keratoconjunctivitis sicca (non-SS KCS) and Sjögren's Syndrome (SS) patients | USA/English | 210 (total) 48 (control) (73) 130 (non-SS KCS) (79) 32 (SS) (91) | 39.2 (control) (20–66) 55 (non-SS KCS) (22–89) | Dry eye symptoms 4 weeks previous the inclusion |
| 5. | Fairchild et al. 2008 [29] | Psychometric evaluation study of dry eye symptoms | Patients with dry eye | USA/English | 100 (70) | 54 (17) | Not specified |
| 6. | Zheng et al. 2017 [30] | Development and psychometric evaluation study (cross cultural validation) | Patients with dry eye | China/ Chinese | 90 (72 | 50 (20–70) | Not specified |
| 7. | Schaumberg et al. 2007 [31] | Repeatability evaluation study of dry eye syndrome | Patients with dry eyes | USA/English | 52 (92) | 60 (50–71) | Not specified |
| 8. | Amparo et al. 2015 [32] | Testing construct validity | Patients with dry eyes | USA/English | 114 (62) | 52 | Not specified |
| 9. | Grubbs et al. 2014 [33] | Examine psychometric properties (IDEEL construct validity) | Patients with dry eye | USA/English | 66 20 (control) (75) 46 (Dry eye) (80 | 63 (13) (control) 61.6 (13.1) (Dry eye) | Not specified |
| 10. | Hwang et al. 2017 [34] | Determining Minimal Clinically Important Difference | Patients with dry eyes | USA/English | 33 (68) | 60.5 | Not specified |
| 11. | Hosseini et al. 2018 [35] | Development and psychometric evaluation study of blepharitis patients | Patients with blepharitis | USA/English and Spanish | 907 (57) | 62 (19–93) | Not specified |
| 12. | Sakane et al. 2013 [36] | Development and psychometric evaluation study | Patient with and without dry eye symptoms | Japan/ Japanese | 224 (total) 21 (86) (control) 203 (94) (Dry eye) | 71 (9) (control) 64 (13) (Dry eye) | Not specified |
| 13. | Paugh et al. 2016 [37] | Development and psychometric evaluation study of Meibomian Gland Dysfunction patients | Patients with dry eye | USA/English | 69 (78) | 53 (18) | Not specified |
| 14. | Nichols et al. 2002 [38] | Reliability evaluation study of dry eye syndrome | Patients with dry eye | USA/English | 75 (71) | 46 (21–81) | Not specified |
| 15. | Johnson et al. 2007 [39] | Examine psychometric properties | Students and staff from Cardiff University and University Hospital of Wales | UK/English | 452 (6) | 34 (18–75) | Not specified |

*(Continued)*

**Table 1.** (Continued)

| | Study | Type of study | Study population and setting | Country/ Language | Sample size (% females) | Mean age in years (SD or range) | Duration of condition/ treatment Months, Mean (SD) unless otherwise stated |
|---|---|---|---|---|---|---|---|
| 16. | Ngo et al. 2013 [40] | Psychometric evaluation study of dry eye symptoms | Patient with and without dry eye symptoms | Canada/ English | 50 (80) | 47 (20–86) | Not specified |
| 17. | Frost et al. 1998 [41] | Development and validation study of dry eye symptoms and aspects of patients' daily life | Patients with visual impairment | UK/English | 287 (total) 38 (58) (Quality of Life interviews) 106 (50) (1st Stage pre-testing) 51 (54) (2nd Stage pre-testing) 92 (56) (Pilot phase) | 77 (50–91) (Quality of Life interviews) 73 (47–91) (1st Stage pre-testing) 70 (46–89) (2nd Stage pre-testing) 72 (41–91) (Pilot phase) | Not specified |

## Ocular Surface Disease Index (OSDI) PROM

The OSDI questionnaire was developed by Allergan Inc. (Irvine, US) based on internal research derived from the experience of patients and clinicians [26]. The original OSDI version was comprised of 40 items covering eye discomfort, visual tasks and environmental triggers then it was shortened to a 12 item version evaluating ocular soreness due to DED and its relationship with visual function [26]. The 12-item OSDI has 3 domains of interest namely: vision-related function (6 questions), ocular symptoms (3 questions) and environmental triggers (3 questions). Score range from 0 to 100 (a higher score indicates a greater impact). The questionnaire was designed to distinguish between healthy patients and patients with DED.

**PROM development and content validity.** The development process and the development of the content for items were not reported in any of the studies.

**Structural validity and internal consistency.** Only 2 out of 3 OSDI papers assessed factor analysis that reflects the structural validity of the construct being measured. Schiffman et al. [26] and Dougherty et al. [27] performed exploratory factor analysis however limited details were provided on the results. Schiffman et al. [26] conducted a confirmatory factor analysis, whereas Dougherty et al. [27] performed Rasch analysis on the OSDI items. OSDI items were considered to have an acceptable fit to the Rasch model and able to discriminate between patients as demonstrated by the person separation index (PSI 2.16). Schiffman et al. [26] did not perform Rasch analysis but reported a Cronbach's alpha of 0.92 which is considered acceptable within the COSMIN guidelines (95% Confidence interval range 0.89–0.94).

**Reliability.** Schiffman et al.[22] evaluated test-retest reliability (reproducibility) and they reported an intraclass correlation (95% Confidence Interval) of 0.82 (range 0.73–0.88) [22].

**Construct validity.** OSDI showed moderate correlation with other considered PROMs such as the McMonnies questionnaire ($r = 0.67$, $p = <0.001$) and the NEI VFQ-25 ($r = -0.77$, $p = <0.001$) [26].

**Responsiveness.** Only Miller et al. [25] calculated the Minimal Clinically Important Difference (MCID) for the overall OSDI score using the Change with the clinician Global Impression (CGI) and the Subject Global Assessment (SGA) as anchors. Both CGI and SGA were found significantly related ($r = -0.3979$ and $r = -0.4200$ with $p<0.001$, respectively).

**Table 2. Characteristics of the included studies.**

| | Instrument name | Description/ Domains | Study language (s) | Summary of instrument | Scoring method/Direction | Range Total score per domain (total for all items) | Mode of administration | Recall period |
|---|---|---|---|---|---|---|---|---|
| 1. | Ocular Surface Disease Index (OSDI) [25] | Assess dry eye symptoms and its effects on vision-related function/3 subscales based on ocular symptoms, vision-related functions and environmental triggers | English (USA) | 12-item instrument | OSDI Likert scale ranging from 0 to 4 points (Score from 0 to 100)/Higher scores means higher impact on QoL | 0–20 1st subscale 0–16 2nd subscale 0–12 3rd subscale (0–100) | Self-administered | 10 to 15 months |
| 2. | Ocular Surface Disease Index (OSDI) [26] | | | | | | | 2 weeks |
| 3. | Ocular Surface Disease Index (OSDI) [27] | | | | | | | 2 weeks |
| 4. | Impact of Dry Eye on Everyday Life (IDEEL) [28] | Assess the impact of dry eye symptoms on everyday life./ 3 different modules: Dry Eye Symptom-Bother (DESB) 20 items Dry Eye Impact on Daily Life (DEIDL) 27 items Dry Eye Treatment Satisfaction (DETS)10 items | English (USA) | 57-item instrument 3 different modules: Dry Eye Symptom-Bother 20 items Dry Eye Impact on Daily Life 27 items Dry Eye Treatment Satisfaction 10 items | 5-point Likert scale for all items except "eyedrop use" that was scored "Yes" or "No" (dichotomous)/DESB higher scores means greater bother DEIDL higher scores means less impact on QoL DETS higher score means greater satisfaction with treatment | All (0–100) | Self-administered | 2 weeks |
| 5. | Impact of Dry Eye on Everyday Life Symptom Bother (IDEEL-SB) [29] | Dry Eye Symptom-Bother (DESB) 20 items | English (USA) | 20-item instrument | Likert scale (Score max 100)/DESB higher scores means greater bother | (0–100) | Self-administered | 1 and 4 weeks |
| 6. | Chinese version of Dry Eye Related Quality of Life (CDERQOL) [30] | Assess the impact of dry eye symptoms on everyday life./ 3 different modules: Dry Eye Symptom-Bother (DESB) 20 items Dry Eye Impact on Daily Life (DEIDL) 27 items Dry Eye Treatment Satisfaction (DETS)10 items | China | 45-item instrument | 5-point Likert scale ranging from "completely disagree" (1) to "completely agree" (5)/DESB higher scores means greater bother DEIDL higher scores means less impact on QoL DETS higher score means greater satisfaction with treatment | All (0–100) | Self-administered | N/A |
| 7. | Symptom Assessment iN Dry Eye (SANDE) [31] | Using visual analog scale to assess frequency and severity of dry eye syndrome./2 different modules: Frequency and severity of symptoms | English (USA) | 2-item instrument | Visual Analog Scale (VAS)/ Frequency: rarely to all the time Severity: very mild to very severe | All (0–100 mm) | Self-administered | 2 and 4 months |
| 8. | Ocular Surface Disease Index (OSDI) and Symptom Assessment in Dry Eye (SANDE) [32] | | English (USA) | OSDI 12-item instrument SANDE 2-item instrument | OSDI Likert scale ranging from 0 to 4 points (Score from 0 to 100) SANDE Visual Analog Scale (VAS) Both focused on severity and frequency of dry eye symptoms / OSDI Higher scores means higher impact on QoL SANDE Frequency: rarely to all the time Severity: very mild to very severe | OSDI 0–20 1st subscale 0–16 2nd subcale 0–12 3rd subscale (0–100)/ SANDE All (0–100 mm) | Both Self-administered | 81 ± 43 days (median 71 days; range 20–283 days) |

*(Continued)*

**Table 2.** (Continued)

| | Instrument name | Description/ Domains | Study language (s) | Summary of instrument | Scoring method/Direction | Range Total score per domain (total for all items) | Mode of administration | Recall period |
|---|---|---|---|---|---|---|---|---|
| 9. | University of North Carolina Dry Eye Management Scale (UNC DEMS) [33] | Using visual analog scale to assess how bad are dry eye symptoms and how they affecting quality of life./ Frequency and severity of symptoms | English (USA) | Single-item instrument | Scale from 1 to 10/ Frequency: not at all to greatly Severity: not a problem to severe | All (0–100 mm) | Not clear how administered | 1 week |
| 10. | University of North Carolina Dry Eye Management Scale (UNC DEMS) [34] | | | | | | Technician or student researchers | 2 weeks |
| 11. | BLepharItIS Symptom (BLISS) measure [35] | Blepharitis symptoms related./Frequency of symptoms | English (USA) and Spanish | 13-item instrument Blepharitis symptoms related | Response range from "none of the time" (0) to "all of the time" (3)/From "none of the time" (0) to "all of the time" (3) | 0 to 39 | Self-administered | 2 weeks |
| 12. | Dry Eye–Related Quality-of-Life Score (DEQS) [36] | Measuring experience of various eye symptoms and how affecting quality of life./ Frequency and severity of symptoms | Japan | 15-item instrument | Likert scale ranging from 0 (no symptom) to 4 (highest frequency of symptom)/ From "never" (0) to "always" (4) and from "hardly bothered me" (0) to "bothered me very much" (4) | 0 to 126 | Self-administered | 2 weeks |
| 13. | Meibomian Gland Dysfunction (MGD) Symptom Questionnaire [37] | Measuring symptoms in MGD sufferers./Frequency and intensity | English (USA) | 14-item instrument | Likert scale/From "normal" (0) to "severe" (9) | 0 to 126 | Self-administered | 1 week |
| 14. | National Eye Institute Visual Function Questionnaire (NEI-VFQ) [38] | Visual function and quality of life related to vision./ General health 1 item Vision-related quality of life 15 items Vision problems 9 items | English (USA) | 25-item instrument | Likert scale (Score from 0 to 100)/ General health from "excellent" (1) to "poor" (5) Vision-related QoL from "no difficult at all" (1) to "stopped doing this for other reasons or not interested in doing this" (6) Vision problems from "all of the time" (1) to "none of the time" (5) | 0 to 100 | Self-administered | 2–3 weeks |
| 15. | Ocular Comfort Index (OCI) [39] | Quick assessment of the ocular comfort./Frequency and intensity over 8 modules ranging from comfort, visual stability, tiredness, stinging, pain, itching, grittiness | English (UK) | 12-item instrument | Scale from 0 to 100 From "no symptoms" (0) to "most frequent/most severe in the past week" (6) | 0 to 72 | Self-administered and returned via mail | 28 days |
| 16. | Standard Patient Evaluation of Eye Dryness (SPEED) [40] | Quick tracking of the progression of dry eye symptoms over time./ Frequency and severity of symptoms over 8 modules dryness, grittiness, scratchiness, irritation, burning, watering, soreness, and eye fatigue | English (Canada) | 8-item instrument | Likert scale ranging from 0 to 3 and from 0 to 4 (question 3) (Score from 0 to 28)/ Frequency From "never" (0) to "constant" (3) Severity From "no problems" (0) to "intolerable" (4) | 0 to 28 | Self-administered | 1 week |

(*Continued*)

**Table 2.** (Continued)

| | Instrument name | Description/ Domains | Study language (s) | Summary of instrument | Scoring method/Direction | Range Total score per domain (total for all items) | Mode of administration | Recall period |
|---|---|---|---|---|---|---|---|---|
| 17. | Vision Quality Of Life core questionnaire (VCM) [41] | Target questionnaire toward vision-related patient perception of quality of life./ Frequency over embarrassment, anger, depression, loneliness, fear of deterioration, in vision, safety at home, safety outside the home, coping with everyday life, inability to do preferred activities and life's interference | English (UK) | 10-item instrument | Scale from 0 (no problem) to 5 (extreme problem)/ From "not at all" (0), "very rarely" (1) "a little of the time" (2) "a fair amount of the time" (3) "a lot of the time" (4) and "all the time" (5) | 0 to 50 | Self-administered + Administered by interviewer in case of severe visual impairment | 7–38 days (median 17 days) |

N/A not available.

## Impact of Dry Eye in Everyday Life (IDEEL)

The Impact of Dry Eye on Everyday Life (IDEEL) questionnaire is a comprehensive dry eye specific questionnaire developed by Alcon Research LTD (France) to evaluate symptom-related bother, impact on daily life and treatment satisfaction in a population with dry eye [28]. There are 57 items grouped into 3 different modules. The "Dry Eye Symptom-Bother" module includes 20 items with a 4-point Likert scale ("not at all", "slightly", "moderately" and "very much") and frequency items score with a 5-point Likert scale. The "Dry Eye Impact on Daily Life" module is composed of 3 modules including Impact due to dry eye on Daily Living Activities (9 items), Emotional Impact (12 items) and Impact on Work (6 items). Daily living and emotional impact are evaluated with a 5-point Likert scale. Work status consists of dichotomous responses (e.g. "Yes" or "No"). A higher the score means less impact on daily tasks, work and emotion. Finally, the "Dry Eye Treatment Satisfaction" module is composed of 10 items which are divided into "Satisfaction with Treatment Effectiveness (6 items)" and "Treatment-Related Bother/Inconvenience (4 items)" all the items are scored with a 5-point Likert scale.

**PROM development and content validity.** Abetz et al. [28] provided a clear description of the constructs being measured and the study population which consisted of dry eye patients [non-Sjögren's (non-SS) and Sjögren's Syndrome (SS)] and a control group (made up of healthy subjects). However, neither Fairchild et al. [29] and Zheng et al. [30] provided enough information to adequately assess the PROM development process. Content validity was only assessed by Abetz et al. [28] where the items were developed based on patients' input (pilot study with 16 patients) and then refined using clinicians opinions.

**Structural validity and internal consistency.** Abetz et al. [28] used exploratory factor analysis (EFA) to identify and remove 55 items. Zheng et al. [30] performed the Kaiser-Meyer-Olkin (KMO) Measure of Sampling Adequacy and the Bartlett test of sphericity before performing EFA. Zheng et al. [30] results obtained demonstrated that the data were suitable to perform EFA (KMO = 0.840: Bartlett = 1527.1, $p < 0.001$). The Rasch analysis was not performed in any of the papers considered. Following the COSMIN guidelines, the sample size considered should be at least 5 times the number of items included in the PROM. The number of items included in the questionnaires analysed by Abetz et al. [28] (n = 57), Fairchild et al.

**Table 3. Results of studies on measurement properties.**

| PROM (ref.) | Country[a] | Structural validity (3) | | | Internal consistency (4) | | | Reliability (6) | | | Hypotheses testing (9) | | | Responsiveness (10) | | |
|---|---|---|---|---|---|---|---|---|---|---|---|---|---|---|---|---|
| | | n | Meth qual. | Result (rating) | n | Meth qual. | Result (rating) | n | Meth qual. | Result (rating) | n | Meth qual. | Result (rating) | n | Meth qual. | Result (rating) |
| 1. OSDI [25] | USA | DNA | DNA | DNA | DNA | DNA | DNA | DNA | DNA | DNA | DNA | DNA | DNA | 310 | adequate | + |
| 2. OSDI [26] | USA | 139 | adequate | ? | 139 | Very good | Cronb. Alpha = >0.70 (+) | 76 | adequate | ICC ≥ 0.70 (+) | 139 | adequate | + | DNA | DNA | DNA |
| 3. OSDI [27] | USA | 172 | v. good | (-) | DNA | DNA | DNA | DNA | DNA | DNA | 172 | adequate | + | DNA | DNA | DNA |
| 4. IDEEL [28] | USA and CANADA | 210 | inadequate | ? | 210 | v. good | Cronb. Alpha = >0.70 (+) | 167 | adequate | ICC ≥ 0.70 (+) | 210 | adequate | + | DNA | DNA | DNA |
| 5. IDEEL-SB [29] | USA | DNA | DNA | DNA | DNA | DNA | DNA | DNA | DNA | DNA | DNA | DNA | DNA | 74 | adequate | + |
| 6. CDERQOL (IDEEL) [30] | CHINA | 90 | inadequate | ? | 90 | v. good | Cronb. Alpha = >0.70 (+) | DNA | DNA | DNA | 90 | adequate | ? | DNA | DNA | DNA |
| 7. SANDE [26] | USA | DNA | DNA | DNA | DNA | DNA | DNA | 52 | inadequate | - | 52 | adequate | + | DNA | DNA | DNA |
| 8. SANDE [32] | USA | DNA | DNA | DNA | DNA | DNA | DNA | DNA | DNA | DNA | 114 | adequate | + | DNA | DNA | DNA |
| 9. UNC DEMS [13] | USA | DNA | DNA | DNA | DNA | DNA | DNA | 57 | doubtful | ? | 66 | adequate | + | 33 | adequate | + |
| 10. UNC DEMS [34] | USA | DNA | DNA | DNA | DNA | DNA | DNA | DNA | DNA | DNA | DNA | DNA | DNA | DNA | DNA | DNA |
| 11. BLISS [35] | USA | 907 | v. good | + CFI 0.954 | DNA | DNA | DNA | 907 | adequate | ? | 907 | adequate | + | DNA | DNA | DNA |
| 12. DEQS [36] | JAPAN | 224 | v. good | ? | 224 | v. good | Cronb. Alpha = >0.70 (+) | 116 | adequate | ICC ≥ 0.70 (+) | 224 | adequate | + | 10 | adequate | + |
| 13. MGD [37] | USA | 69 | doubtful | + | DNA | DNA | DNA | DNA | DNA | DNA | 69 | adequate | ? | 69 | adequate | ? |
| 14. NEI-VFQ [38] | UK | DNA | DNA | DNA | DNA | DNA | DNA | 75 | doubtful | weighted Kappa < 0.70 (-) | DNA | DNA | DNA | DNA | DNA | DNA |
| 15. OCI [39] | UK | 452 | v. good | + | DNA | V. good | Cronb. Alpha not reported PSI 2.66 and ISI item 11.5 | 95 | adequate | ICC ≥ 0.70 (+) | 337 | adequate | + | 452 | adequate | ? |
| 16. SPEED [40] | CANADA | 50 | doubtful | ? | DNA | DNA | DNA | 50 | adequate | ? | 50 | adequate | + | DNA | DNA | DNA |
| 17. VCM [41] | UK | 184 | adequate | ? | 184 | doubtful | Cronb. Alpha = >0.70 (+) | 26 | doubtful | ICC <0.70 | 40 | adequate | + | DNA | DNA | DNA |

DNA = Data not available for assessment.

PSI = Person separation index.

ISI = item separation index.

**Table 4. Updated COSMIN criteria for good measurement properties.**

| Measurement property | Rating | Criteria |
|---|---|---|
| Structural validity | + | **CTT:**<br>CFA: CFI or TLI or comparable measure >0.95 OR RMSEA <0.06 OR SRMR <0.08<br>**IRT/Rasch:**<br>No violation of unidimensionality: CFI or TLI or comparable measure >0.95 OR RMSEA <0.06 OR SRMR <0.08<br>*AND*<br>no violation of <u>local independence</u>: residual correlations among the items after controlling for the dominant factor < 0.20 OR Q3's < 0.37<br>*AND*<br>no violation of <u>monotonicity</u>: adequate looking graphs OR item scalability >0.30<br>*AND*<br>adequate <u>model fit</u>:<br>IRT: $\chi 2$ >0.01<br>Rasch: infit and outfit mean squares $\geq$ 0.5 and $\leq$ 1.5 OR Z-standardized values > -2 and <2 |
| | ? | CTT: Not all information for '+' reported IRT/Rasch: Model fit not reported |
| | – | Criteria for '+' not met |
| Internal consistency | + | At least low evidence for sufficient structural validity AND Cronbach's alpha(s) $\geq$ 0.70 for each unidimensional scale or subscale |
| | ? | Criteria for "At least low evidence for sufficient structural validity" not met |
| | – | At least low evidence for sufficient structural validity AND Cronbach's alpha(s) < 0.70 for each unidimensional scale or subscale |
| Reliability | + | ICC or weighted Kappa $\geq$ 0.70 |
| | ? | ICC or weighted Kappa not reported |
| | – | ICC or weighted Kappa < 0.70 |
| Measurement error | + | SDC or LoA < MIC5 |
| | ? | MIC not defined |
| | – | SDC or LoA > MIC5 |
| Construct validity | + | The result is in accordance with the hypothesis |
| | ? | No hypothesis defined (by the review team) |
| | – | The result is not in accordance with the hypothesis |
| Cross-cultural validity \measurement invariance | + | No important differences found between group factors (such as age, gender, language) in multiple group factor analysis OR no important DIF for group factors (McFadden's R2 < 0.02) |
| | ? | No multiple group factor analysis OR DIF analysis performed |
| | – | Important differences between group factors OR DIF was found |
| Criterion validity | + | Correlation with gold standard $\geq$ 0.70 OR Area Under Curve (AUC) $\geq$ 0.70 |
| | ? | Not all information for '+' reported |
| | – | Correlation with gold standard < 0.70 OR AUC < 0.70 |
| Responsiveness | + | The result is in accordance with the hypothesis OR AUC $\geq$ 0.70 |
| | ? | No hypothesis defined (by the review team) |
| | – | The result is not in accordance with the hypothesis OR AUC < 0.70 |

The criteria are based on e.g. Terwee et al. [42] and Prinsen et al. [43] (Reproduced with kind permission from Caroline Terwee, COSMIN).

[29] (n = 20) and Zheng et al.[30] (n = 45) none of the study cohorts considered were numerically large enough to satisfy this requirement. Abetz et al. [28] tested internal consistency and obtained a Cronbach's alpha ($\geq$ 0.70) that ranged from good (impact on daily activities, impact

on work and satisfaction with the treatment effectiveness) to excellent ("Dry Eye Symptom-Bother" module). Zheng et al. [30] confirmed internal consistency across all 5 domains with Cronbach's alpha's > 0.70 on the original IDEEL. It is not clear if Fairchild et al. tested for internal consistency [29].

**Reliability.** Abetz et al. [28] demonstrated test-retest reliability using intraclass correlation coefficient (ICC) ranging from 0.70 to 0.88 over the two time points. Neither Fairchild [29] nor Zheng [30] papers reported information on reliability testing.

**Construct validity.** Abetz et al. paper [28] demonstrated low correlations for IDEEL with two common and generic quality of life (QoL) questionnaires (Short Form 36-item and the EuroQol (EQ) 5-item but was more highly correlated with the Dry Eye Questionnaire (DEQ) on items related to eye dryness and eye discomfort (ranging from 0.21 to 0.83).

Zheng et al. [30] assessed known group validity by performing pairwise comparisons for domains such as "Dry Eye Symptoms Bother" and "Impact on Daily Activities" reporting a significant difference between patients belong to the mild (n = 24) and moderate (n = 36) DED groups from those within the severe group (n = 30) (p< 0.05).

**Responsiveness.** Fairchild et al. [29] determined responsiveness by assessing changes in self-assessed severity in patients with DED following the administration of eye drops for 4 weeks. The authors also used a response-operator curve (ROC) to determine clinically important differences for the IDEEL"Symptom Bother" module. The clinically important difference (CID) value which has minimised the overall error (CID at <12, 68% agreement, K = 0.34, Effect size 1.14). After 4 weeks of drops usage, IDEEL-SB dropped among "improved" subjects by -13.3 (SD = 10.9), "same" shifted by -4.7 (SD = 9.4), "worsened" changed by 1.4 (SD = 11.1).

## Symptom Assessment iN Dry Eye (SANDE)

The SANDE questionnaire is composed of two questions determine how eye dryness and/or irritation impact on patient symptomatology using a visual analogue scale (VAS) [31]. The scale lengths 100mm, determines averages of frequency and severity symptoms for patients having ocular discomfort or dryness. In version 1 of the questionnaire, the patient is asked to mark on two given lines (frequency/severity) a lower score (left of the VAS) accounts for "rarely/very mild" symptoms and a high score (right of the VAS) "all of the time/very severe" symptoms. A second refined version of the questionnaire was created on which an anchor is placed in the middle of each line (frequency/severity) to orient the patient on reporting based on the last visit received [31]. On both lines, the extreme left accounts for "rarely"/"very mild" and the extreme right for "all of the time"/"very severe" according to the change perceived from the previous visit.

**Reliability.** Schaumberg et al. [31] reported a low ICC when comparing SANDE scores before and after 2 months (range 0.12 to 0.39) while increased ICC values (range 0.45 to 0.76) were reported when SANDE scores were compared closer (questionnaire obtained during visit vs questionnaire emailed two days after the visit).

**Construct validity.** Schaumberg et al. [31] compared the SANDE questionnaire with the degree of corneal staining and the reported use of eyedrops within all the consultations recorded (initial, 2-month and 4-month follow-up): a weak correlation was found with the clinical test (range 0.04 to 0.15) while a strong correlation was found considering the treatment used (range 0.43 to 0.50). Amparo et al. [32] normalised the original scores when comparing SANDE with OSDI as both questionnaire measure the symptomatology in a different way. The authors obtained a significant correlation when the questionnaires were compared at the baseline (r = 0.64, p< 0.0001) remarked also when considering mild to moderate dry eye

patients (r = 0.37, p = 0.045) and severe dry eye patients (r = 0.39, p< 0.0001). Overall, both questionnaires were correlated in terms of severity and frequency (r = 0.60, p = 0.0001) scores.

### University of North Carolina Dry Eye Management Scale (UNC DEMS)

The UNC DEMS questionnaire was developed following the Patient-Reported Outcome Measurement Information System (PROMIS®) guidelines [44]. UCM DEMS is a one-item graded scale from 1–10 developed by ophthalmologists and people with DED. The questionnaire works asking patients their dry eye symptoms which can include *pain*, *burning*, *tearing*, *grittiness*, "feeling like something is in your eye", and/or *sensitivity* to light. The symptoms bothersome increase from the extreme left (1) up to the extreme right (10) based over the past week [33].

**PROM development and content validity.** Grubbs et al. [33] used the first seven PROMIS® standards for developing the questionnaire [44]. The authors clearly described the construct by starting with a deep review of the literature on the field (DED symptoms and its influence on QoL) and also added discussions with clinicians and patients suffering from the conditions. In details, the development has considered an initial pilot study where 18 patients with DED were asked to complete the questionnaire followed by a 15-minute cognitive interview based on a question template of 13 items. Subjects enrolled in the study were able to express their opinions on the comprehensiveness of the PROM. However, it is not clear from the article if the interviewers were trained or not.

**Reliability.** Grubbs et al.[33] determined the re-test reliability of the UNC DEMS using 50 patients. The ICC was 0.90 (range 0.84–0.95) [26].

**Construct validity.** The comparison done by Grubbs et al. [16] with the OSDI questionnaire reported a significant correlation coefficient (Pearson r = 0.80, p<0.001). Although, when compared inside the DED group enrolled the coefficient decreased to 0.69 but still significant (p<0.001).

**Responsiveness.** Hwang et al. [34] considered two methods to estimate minimal clinically important difference (MCID). The linear regression was adjusted for both the number of days since the last visit and the previous UNC DEMS score. The adjusted regression yielded a beta coefficient value of -0.56; a confidence interval ranging from -0.99 to -0.13 ($R^2$ = 0.43, p = 0.013). The anchor method yielded an average change in the score for those rating their symptom change to be "a little better/worse" was 1.09 (n = 11).

**BLepharItIS Symptom (BLISS) measure.** The study population was composed of patients with blepharitis (e.g. anterior inflammation of the eyelid margins) [35].

The BLISS questionnaire is made of 13 items that assess symptoms related to ocular discomfort and eye irritation caused by blepharitis (e.g. inflammation of the margin of the eyelids) [35]. All the presented items have been through an FDA Division of Transplant and Ophthalmology Products review (more information can be found: https://www.fda.gov/about-fda/center-drug-evaluation-and-research-cder/office-infectious-diseases-oid).

**PROM development and content validity.** The FDA review suggested that BLISS time setting for the questions should be set as "today". However, any additional information were given on the content validity and its assessment.

**Structural validity and internal consistency.** Hosseini et al.[35] performed both exploratory and confirmatory factor analysis for all the items considered using PRO-MAX rotation therefore confirmed considering categorical factor analysis using Mplus (Muthen & Muther, Los Angeles, California, US). Sample size considered was very good with 907 subjects enrolled with a clinical diagnosis of blepharitis.

**Reliability.** The "irritation scale" and "debris scale" returned a Cronbach alpha of 0.88 and 0.85 respectively. Test-retest reliability done with Spearman rank-order correlations

ranged from 0.58 for "eyes that itch" to 0.74 for "dry eyes" when screening and beginning (7 days later) of the treatment visits were compared. These values represent moderate to good reliability [45].

**Construct validity.**   Spearman rank-order correlations between BLISS and OSDI were significant with 0.63 and 0.41 for both "irritation" and "debris" scale versus 12-item OSDI (p's< 0.001) [35].

**Responsiveness.**   None of the included studies provided information on any of these measurement properties.

## Dry Eye-Related Quality-of-Life Score (DEQS) questionnaire

DEQS questionnaire is used for assessing the severity of dry-eye associated symptoms using a score from 0 to 100, where 0 indicates no bother while 100 indicates a higher impact of symptoms on daily life [36]. DEQS is composed of 15 items and 2 subscales that are the "Bothersome Ocular Symptoms" and "Impact on Daily Life". These are scored using a 2-step approach based on frequency and severity of the condition. The frequency is scored with a 5-point Likert scale ranging from 0 to 4 where 0 corresponds to no symptoms experienced and 4 highest frequency of symptoms). Severity is scored using a 4-point Likert scale ranging from 1 to 4 where 4 means the highest severity.

**PROM development and content validity.**   The initial DEQS had 45 items which were reduced to 35 items after consulting with patients with DED and expert clinicians on the field. Participants were interviewed regarding their general impression, comprehensiveness, clarity of the instructions, readability of the format and layout and opinions gathered from reading the questionnaire.

**Structural validity and internal consistency.**   A preliminary study was performed using factor analysis. The results returned the presence of a 2-step scales known as "Bothersome Ocular Symptoms" (6 items) and "Impact on Daily Life" (9 items). Internal consistency demonstrated Cronbach's alphas for "Impact on Daily Life" and "Bothersome Ocular Symptoms" with a summary score of the two scales of 0.93 [36].

**Reliability.**   Reliability of the DEQS was determined via test-retest in 116 DED subjects who have completed the questionnaire between 8 to 21 days after the first sample was taken. ICC correlation coefficients ranged from 0.81 to 0.93 [36].

**Construct validity.**   Through known group comparison, the validity of the DEQS was tested comparing DED subject with a control group of no-DED subjects. All DEQS items ("Bothersome Ocular Symptom", "Impact on Daily Life" and summary score) were significantly correlated with the 25-item Visual Function Questionnaire (VFQ-25) with values ranged from -0.20 to -0.77. Also, DEQS significantly correlated with the 8-item Short Form (SF-8) questionnaire with the correlations ranged from -0.27 to -0.52 [36].

**Responsiveness.**   Responsiveness was tested only in 10 DED subjects with punctal plug insertion. The results revealed an improvement recorded with DEQS after the treatment considering the "Impact on Daily Life" (before $37.2 \pm 27.7$ vs after $20.7 \pm 25.1$, p = 0.04), "Bothersome Ocular Symptoms" (before $49.6 \pm 16.0$ vs after $19.3 \pm 13.0$, p = <0.001) and summary score (before $42.1 \pm 21.6$ vs after $20.0 \pm 19.0$, p = 0.001).

## Meibomian gland dysfunction (MGD)-specific questionnaire

MGD condition is one of the main reason for developing DED. MGD-specific questionnaire is composed of 24 items using a 0 to 9 scale for frequency and intensity. Additionally, the last question is an open-ended question that read as "What was the most significant symptom you have experienced with your eyes in the past month?" [37].

**PROM development and content validity.**   The development of the questionnaire used a psychometric approach where two major surveys were considered to determine the number of scale steps and its progression. The final version of the questionnaire has considered an ordinal scale including patient's symptoms during the last month (frequency and intensity).

**Structural validity and internal consistency.**   Rasch analysis was performed in three different steps. Initially on all 24 items, then by reducing items first to 18 items and finally a 14-items questionnaire. Including 69 subjects and this demonstrated reasonable mean square levels (INFIT MNSQ = 0.97; ZSTD = 20.2; OUTFIT MNSQ = 0.96; ZSTD = 20.2).

**Reliability.**   Test-retest reliability was not performed.

**Construct validity.**   The comparison between MGD-specific questionnaire and Schein questionnaire (previously used in MGD patients [36]) revealed a significant Pearson correlation at the baseline (r = 0.71, p<0.001) and at 6 months (r = 0.76, p<0.001).

**Responsiveness.**   The authors measured the responsiveness of the questionnaire by administrating it before and after 6-month of an eye drop treatment (mid-viscosity artificial tears). MGD-specific questionnaire reported significant changes over the treatment: 58.4 ± 29.3 and 56.8 ± 22.3 (baseline), 30.6 ± 26.8 and 26.6 ± 24.4 (6 months) for the propylene glycol–HP-guar and carboxyl-methyl-cellulose 0.5% treatments respectively (p = 0,001) [37].

## National Eye Institute Visual Function Questionnaire 25-item (NEI-VFQ-25)

NEI-VFQ-25 is a questionnaire designed to evaluate the visual function and its related QoL [38]. It includes different areas based around visual function and wellbeing and also two items to detect the level of patients tolerance to ocular irritation [38]. The total number of the items included are 25 with a scoring system from 0 to 100 (a lower score indicates a greater impact).

**Reliability.**   The authors assessed test-retest reliability over a short interval (e.g. between 1 and 2 weeks). They used kappa values above 0.60 to evaluate responsiveness items related to "difficulty driving at night" and "staying at home because of vision" reported kappa values of 0.85 and 0.90 respectively. ICC coefficients ranged from 0.57 to 0.88 [38].

## Ocular Comfort Index (OCI) questionnaire

The OCI questionnaire is composed of 12 items which measure not only ocular surface irritation but also the impact of DED on patients' wellbeing and how effective are the treatment considered. It is used in clinical trials [39].

**PROM development and content validity.**   Development of the PROM started with a literature review and patient interviews.. However, it was not stated whether cognitive de-briefing interviews took place. Given the limited information we were unable to assess whether the content validity of the OCI met with the COSMIN criteria. Areas considered were: comfort, dryness, grittiness, itching, pain, stinging, tiredness and visual stability that were focused on 15 questions. Except for comfort, all the areas included were assessed for frequency and intensity. Ten versions of the OCI were produced and tested with the items in random order (except comfort).

**Structural validity and internal consistency.**   Unidimensionality was tested using unrotated factor analysis and fit to the Rasch model. The principal factor correlated with the individual items ranging from 0.63 to 0.79. Additionally, Rasch analysis was performed using the Rasch analysis software Winsteps (Winsteps, Chicago, IL, US) After the first Rasch analysis was performed and 3 misfitting items were removed, the reduced 12-item scale showed good fit. Fit statistics included INFIT MNSQ ranging from 0.86 to 1.17 and ZSTD from -2.0 to 2.4. OUTFIT MNSQ ranging from 0.75 to 1.20 and ZSTD from 0.61 to 0.73. The separation indices

in the OCI demonstrated a good level of person/item separation with a person separation of 2.66 indicating stable item difficulty estimates and discrimination between 3 levels of ability across the scale [46] and an item separation of 11.12.

**Reliability.** Test-retest was performed to assess test reliability. 100 subjects repeated the questionnaire 14 ± 7 days the authors reported 95% CI for the two-way random effects ICC of the OCI was between 0.81 and 0.91.

**Construct validity.** OCI demonstrated significant correlations with the OSDI questionnaire ranging from 0.68 to 0.78 (p< 0.0001).

**Responsiveness.** OCI demonstrated that it could identify changes in symptoms before and after eye drop treatments (28 days with either 0.3% carbomer 934 eyedrops or 0.18% sodium hyaluronate eye drops with 65 subjects): 95% CI of the treatment difference accounted for -5.5 to -8.0 units (p< 0.0001; paired t-test) [39].

## Standard Patient Evaluation of Eye Dryness (SPEED) questionnaire

SPEED questionnaire was developed to measure symptoms of ocular dryness and how they change over time [40]. It is composed of 8 items and the final score range from 0 to 28.

**Structural validity and internal consistency.** Ngo et al. [40] applied Rasch analysis to determine if the SPEED questionnaire fitted requirements of person and item fit requirements for unidimensionality. The authors claimed that infit and outfit statistics met the requirement for fit to the Rasch model but neither statistics were provided to support this claim. Besides, the reported sample size was small (n = 50), and did not reach the suggested COSMIN requirements for adequate sample size in relationship to the number of SPEED items, and therefore the likelihood of identifying misfit is limited should it occur.

**Reliability.** Test-retest was performed 1 week apart and the concordance correlation coefficient (CCC) was 0.923 (95% confidence interval, 0.868–0.955) (where 1.0 is considered perfectly concordant test-retest data).

**Construct validity.** The area under the receiver-operator curves (AUC) comparing SPEED and OSDI on 1 day after and 1 week after, was 0.928.

## Vision Core Measure1 (VCM1)

VCM1, a vision-related quality of life (VQoL), is composed of 10 items that evaluate symptomatology and feeling of patients with visual impairment [41].

**PROM development and content validity.** Interviews with 38 visual impaired adults and with 37 professionals and support workers were considered to generate the relevant items for the questionnaire. Additionally, a literature review was performed. Content validity was assessed together with face validity in 184 subjects with different visual limitations and social backgrounds. The results gathered confirmed the high level of content validity [41].

**Structural validity and internal consistency.** Structural validity was not assed while the internal consistency showed a Cronbach alpha coefficient of 0.93 for the 10 items selected (pre-testing phase and pilot phase) [41].

**Reliability.** A pilot study with 92 subjects was conducted to determine reliability: test-retest was performed between 7 and 38 days (median 17 days). The score ranged from 0.0 to 3.5 where the mean change in VCM1 score was +0.03 with a 95% confidence interval of -0.10 to 0.7.

**Construct validity.** Construct validity was assessed in 40 individuals. VCM1 score correlated with all 129 items in the parent questionnaire and also the correlations with vision-specific measures were generally high that the one observed with the generic measures (Spearman correlations equal to 0.60 or greater) [41].

## Discussion

Different PROMs are available in the field of ophthalmology and visual science [47]. However, it is not completely clear which criteria have been considered when determining the quality of studies evaluating PROMs for use in research and clinical practice. This is the first systematic review to use the Consensus-based Standards for the selection of health Measurement INstruments (COSMIN) Risk of Bias checklist in the assessment of the measurement properties of PROMs used specifically in patients with or without DED and OSD.

The systematic review included 17 studies assessing 11 different PROMs used specifically in this group of patients. The PROMs were used to understand the impact of the conditions (DED, MGD or Blepharitis) on patients' QoL in terms of severity and frequency of symptoms or the effectiveness of treatments.

However, the use of the COSMIN Risk of Bias checklist [24], highlighted the fact that measurement properties such as content validity, measurement error and structural validity were not assessed for many of the PROMs included in this review. The development of many of them was also not adequately reported making it difficult to assess the quality of these measures. Additionally, there was limited evidence of patient and public (PPI) involvement in the development of the PROMs included in this review. PPI is essential in the development of PROMs in order to make research more relevant and accessible to a wider public. As previously described by Wilson [12], patients input is crucial to provide better insights on how the disease can affect QoL but also discuss which outcomes could be considered in research. This can be achieved by considering large-scale surveys where patients' experience can be compared before and after treatment (e.g. cancer) or across different patients' perspective. Therefore, as suggested by Selby and Velikova [48], PPI should be present as a core feature in PROM design and application.

Many of the studies were also based on reviews using classical test theory and this is reflected in the COSMIN guidelines where common measurement standards (e.g. validity and reliability are evaluated using factor analysis and Cronbach's Alpha). For example, while the content validity of the OCI could not be determined conclusively, it has good measurement properties and may be used for studies after initial cognitive work is done with patients. However, increasingly modern psychometric approaches such as IRT and Rasch analysis are being used to evaluate the extent to which interval level measurement and unidimensionality is being achieved [49]. Many Rasch papers do not necessarily present CTT statistics as well as IRT or Rasch indices.

As more studies are using only Rasch based methods for evaluating PROMs it might be necessary to have additional guidance within the COSMIN guidelines to reflect this. This approach especially in the development of new PROMs can evaluate additional psychometric properties such as the evaluation of the scoring responses for disordered thresholds.

Previously, Grubbs et al. [17] reported that OSDI and IDEEL questionnaires were two of the most common disease-specific PROMs in the ophthalmic fields, which were considered valid and reliable. The OSDI questionnaire has been available and widely distributed for decades [26]. It is well-known for its brevity and simplicity. However, we could not find any detailed reports on its development which made it difficult to assess the extent to which its content validity is supported by patients' experiences of living with dry eye disease. However, at the time OSDI was developed guidelines on the development of PROMs and in particular making sure people living with the health condition are involved in the development was not common. Content validity using patients' experiences of living with a health condition is considered an important aspect of ensuring PROMs have content validity and this has been recognised by the FDA and EMA [50]. Therefore, we would recommend that further research is

required to ensure that the OSDI reflects patients' experiences of living with dry eye symptoms.

Based on our assessments using the COSMIN checklist, the IDEEL study presented with the highest quality scores in terms of the evaluation of its psychometric properties [51]. They also comply with current FDA guidelines (S2 Table) requiring PRO being used in labelling claims to ensure that they can demonstrate content validity with input from patients lived experiences of the health condition.

As noted in the review by Aiyegbusi et al, these COSMIN standards were developed within the last decade, it is therefore understandable that most of the earlier studies included in our review fared relatively poorly when judged against these exacting standards. This highlights the need for a re-validation of these existing PROMs based on the current methodological standards and recommendations.

## Strength and limitations

This study is the first to undertake a systematic review of PROMs used in DED and OSD, in accordance with the PRISMA [52] and COSMIN guidelines [24]. Following those scientific approaches, has allowed us to conduct a structured and comprehensive evaluation of the measures. However, many of the PROMs considered were unable to fulfil the requirements and the level of detailed reporting was found to be insufficient (e.g. missing data). Another critical aspect observed was related to the populations considered which might be not appropriate for the COSMIN guidelines evaluation (Number of items vs Sample size). However, as reported by Isa et al. [53], COSMIN guidelines might show limitations when judging the methodological quality of uncommon conditions (e.g. Blepharitis) where recruitment is limited because of the rarity of the condition. Also, COSMIN guidelines while are assessing the quality of the PROM's measurement properties does not accommodate Rasch analysis and this should be seen as a limitation too.

## Conclusion

Using the COSMIN guidance it would appear that there are areas for improvement in the study designs evaluating many of the PROMs currently being used in dry eye and ocular surface disease. The majority of the studies included in this review did not meet the proposed criteria and further validation work is required. Using quality PROMs in DED and OSD patients could offer a unique perspective and provide valuable insights into patients' experiences in research and clinical settings. Consequently, it is not always possible to establish the quality of the PROMs being evaluated and future studies should use COSMIN standards as a guide for reporting the evaluation of their psychometric properties.

## Future research

This systematic review could guide future PROMs research in the field of DED and OSD. Researchers may design validation studies to address the gaps in evidence for all the PROMs included in this review. Further evidence on the content validity of the measures including the OSDI is needed.

## Supporting information

**S1 Fig. Systematic review PRISMA 2009 flow diagram.**
(DOCX)

**S1 Table. COSMIN definitions of domains, measurement properties, and aspects of measurement properties.**
(DOCX)

**S2 Table. S2 Table summary of FDA PRO requirements.**
(DOCX)

**S3 Table. Search strategy proms in dry eye disease and ocular surface disease.**
(DOCX)

**S1 File. PRISMA 2009 checklist.**
(DOCX)

## Acknowledgments

The authors thank Dr. Sarah Hughes for her additional statistical guidance in *Winsteps* and for her valuable suggestions for this manuscript.

## Disclaimers

The views expressed in this article are those of the author(s) and not necessarily those of the National Institute for Health Research or the Department of Health and Social Care.

## Author Contributions

**Conceptualization:** Alberto Recchioni, Olalekan Lee Aiyegbusi, Saaeha Rauz, Anita Slade.

**Data curation:** Alberto Recchioni, Olalekan Lee Aiyegbusi.

**Formal analysis:** Alberto Recchioni, Olalekan Lee Aiyegbusi.

**Funding acquisition:** Saaeha Rauz, Anita Slade.

**Investigation:** Alberto Recchioni, Olalekan Lee Aiyegbusi.

**Methodology:** Alberto Recchioni, Olalekan Lee Aiyegbusi, Samantha Cruz-Rivera, Anita Slade.

**Project administration:** Saaeha Rauz.

**Resources:** Alberto Recchioni.

**Validation:** Samantha Cruz-Rivera, Saaeha Rauz, Anita Slade.

**Writing – original draft:** Alberto Recchioni, Olalekan Lee Aiyegbusi.

**Writing – review & editing:** Alberto Recchioni, Olalekan Lee Aiyegbusi, Saaeha Rauz, Anita Slade.

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
