## [Decision Letter · Decision Letter 0]

9 Apr 2021

PONE-D-20-39488

A systematic review assessing the quality of patient reported outcomes measures in dry eye diseases

PLOS ONE

Dear Dr. Rauz,

Thank you for submitting your manuscript to PLOS ONE. After careful consideration, we feel that it has merit but does not fully meet PLOS ONE’s publication criteria as it currently stands. Therefore, we invite you to submit a revised version of the manuscript that addresses the points raised during the review process.

The manuscript has been reviewed by two reviewers. Both acknowledge the relevance of the study and the appropriateness of the analyses. However, both also note several relevant issues of concern and have provided excellent suggestions for improvements. I strongly encourage the authors to take all comments into account and to provide the required clarifications and adjustments to the manuscript. The authors should take particular care to address the issue that both reviewers noted with respect to some of the conclusions drawn and that all conclusions should be clearly based on the findings from this study.

We look forward to receiving your revised manuscript.

Kind regards,

Peter M. ten Klooster, Ph.D.

Academic Editor

PLOS ONE

Journal Requirements:

2. Please include your tables as part of your main manuscript and remove the individual files. Please note that supplementary tables should remain uploaded as separate "supporting information" files.

Reviewers' comments:

Reviewer's Responses to Questions

**Comments to the Author**

1. Is the manuscript technically sound, and do the data support the conclusions?

Reviewer #1: Yes

Reviewer #2: Partly

2. Has the statistical analysis been performed appropriately and rigorously? 

Reviewer #1: Yes

Reviewer #2: Yes

3. Have the authors made all data underlying the findings in their manuscript fully available?

Reviewer #1: Yes

Reviewer #2: Yes

4. Is the manuscript presented in an intelligible fashion and written in standard English?

Reviewer #1: Yes

Reviewer #2: Yes

5. Review Comments to the Author

Reviewer #1: This study seeks to perform a formal search to identify patient-reported outcome measures in dry eye and ocular surface disease and to determine whether these instruments were developed using the COSMIN guidelines or what sort of reporting has been done in other studies for these various instruments. This is an important topic, as PROMs are really useful in dry eye and the evaluation of the quality of the instruments frequently used often seems lacking. Using the COSMIN standard to for evaluation here seems like a good idea.

The abstract background states that the authors “evaluated the psychometric properties of studies” that considered PROMs, but I don’t think that’s quite right. The PROMs have psychometric properties, not the studies, right?

The authors tell us in 110-115 that a previous study by Grubbs on dry eye questionnaires identified the OSDI and IDEEL as the most reliable measure. They note that they think the omission of the term “ocular surface disease” in searching for instruments in that study was a flaw, though they come to similar conclusions in the end. They also note that Grubbs et al. didn’t use the COSMIN guidelines, which they state is the gold standard for this type of work. I would note that the only reference supporting this gold standard statement is by the authors of the COSMIN guidelines. In general, some more discussion of how exactly the COSMIN guidelines are superior to other methods of evaluation would be useful.

I believe the authors could be considerably clearer about how they come to the conclusion that the IDEEL is the only instrument that meets the COSMIN standard. This is not an easy task, perhaps, considering that there are many COSMIN indicators and many different PROMs being considered. However, the reader needs to be able to understand what aspects of the methodological quality of studies on these instruments were judged inadequate. I don’t think the Results section really accomplishes this. For instance, my reading of the section on the Ocular Comfort Index does not turn up any obvious discussion of a problem. When I examine Table 3, again, nothing is particularly obvious.

At line 177 the authors discuss the fact that an overall score is determined by taking the lowest rating of any box considered. Perhaps that score could be reported for each? Sticking with the OCI, in Table 3 I see N/A reported for internal consistency and a question mark for responsiveness (though in the Results text there is a responsiveness study described). Now, in the Structural Validity section of Table 3 for the three IDEEL studies, I see N/As, inadequates, or question marks. Was it the question mark rating that made that aspect acceptable? There must be a way to make the summary of the evaluation more easily accessible to the reader.

Is the N/A for internal consistency the low score for the OCI? Those authors used Winsteps to conduct their Rasch analysis. They reported Rasch person reliability rather than Cronbach Alpha, as is typical in a Rasch study, but could’ve easily just reported the KR-20 value from the same Winsteps output table. If the authors of the OCI study had just reported Cronbach Alpha would that study have a good score for internal consistency? Actually, my reading of the COSMIN reference suggests that the Rasch stats presented in the OCI study should qualify as a measure of internal consistency: “For IRT-based scores: calculate standard error of theta (SE (θ)) or reliability coefficient of estimated latent trait value (index of (subject or item) separation) for each unidimensional scale or subscale”. So perhaps I am misunderstanding what N/A means in this table, but I would’ve rated the OCI study “very good” on internal consistency based on that quote from the COSMIN checklist reference. Could the authors clarify please?

The authors of the COSMIN guidelines have stated: “Note that the COSMIN checklist is not a checklist for the evaluation of the quality of a HR-PRO, but for the methodological quality of studies on their measurement properties.” My opinion is that the language used in this paper is not necessarily consistent with that statement. The final conclusion states that “the IDEEL appears to be the most robust tool and should be considered for use in clinical practice and future research.” The discussion says “…the IDEEL presents the highest quality scores in terms of psychometric properties.”

Actually, the authors are quite fair about noting that many of the instruments evaluated were developed prior to the existence of COSMIN, and in many places are clear that the evaluation is of the studies on the instruments and not the instruments themselves. But it is very easy to slip into a discussion of which PROM is best, rather than which PROM has been evaluated in the best way.

When the guidelines are specifically meant to evaluate the methodological quality of the STUDIES of these instruments, and the lowest score in any of the many evaluation categories is the determining factor, and the guidelines were developed after many of the instruments were, it is not clear to me that statements declaring the IDEEL as the only instrument that should be considered for use in clinic and research are particularly useful. Also, the actual content of the questions of the various surveys may or may not be suited for any individual study or clinical purpose. 116: patients missing an apostrophe

How many times were discrepancies between reviewers noted and how exactly were they resolved?

184-5: What does this sentence mean, exactly? Perhaps it would be useful to note which important measurement properties have not been evaluated for each instrument?

When the 127 records were assessed for eligibility, most were excluded as being out of scope. But there was a scope question in the previous step, in which thousands of records were eliminated. Were there differences in the evaluation of scope at these two stages? I would imagine that the 127 records, having made it through the previous round of eliminations, must’ve been fairly closely related to the target scope.

There is some discussion of Rasch analysis in the paper, but no coverage of the differences between the Classical Test Theory and Item Response Theory approaches and how those differences are relevant to the entire COSMIN approach. I believe this would be beneficial for readers.

Reviewer #2: Dear authors,

PLOS ONE

Manuscript Number: PONE-D-20-39488

Title: A systematic review assessing the quality of patient reported outcomes measures in dry eye diseases

GENERAL CONSIDERATIONS

The authors have proposed to comprehensively evaluate the psychometric properties of studies that considered the use of PROMs in DED and OSD.

- Data is very interesting, however, some concepts and structural changes must be considered

- There rationale of the question of this systematic review must be clear for the readers. The purpose of the abstract seems to be different from the one at the end of the INTRODUCTION.

ABSTRACT

- The authors have mentioned: “comprehensively evaluated the psychometric properties of studies that considered the use of PROMs in DED and OSD.” The purpose of the abstract seems to be different than the one mentioned at the end of the INTRODUCTION. Please rewrite the purpose of the abstract according to the real purpose of the article.

- The authors have mentioned as conclusion: “We suggest considering the IDEEL questionnaire, which showed to be the most robust tool in DED and OSD” Please answer the question proposed in the purpose. Suggesting that the questionnaire is robust was never an issue here. If it is, please change the rationale, sentence in the ABSTRACT and introduction. I would strongly suggest deleting this sentence or citing it in the discussion.

- The authors have mentioned as conclusion: “The majority of the PROMs included in this review did not meet the proposed criteria and further validation work is required.” This answers the purpose of the study mentioned at the end of the INTRODUCTION.

- The authors have mentioned as conclusion: “PROMs in DED and OSD patients could offer a unique perspective and provide valuable insights in research and clinical settings to improve DED patient care.” Please understand that this is not a conclusion so I strongly suggest deleting it or citing it in the DISCUSSION.

DISCUSSION

- The authors have mentioned: “we present a systematic review of studies that report an evaluation of psychometric properties of PROMs developed for use in patients with dry eyes and ocular surface diseases following the latest COSMIN guidelines [18].”

- The authors have mentioned as conclusion: “The quality of different studies evaluating PROMs being used to evaluate the impact of DED and its treatment in patients QoL were reviewed against the exacting COSMIN standards. The majority of those included in this review did not meet the proposed criteria and further validation work is required. PROMs in DED and OSD patients could offer a unique perspective and provide valuable insights in research and clinical settings to improve DED and OSD patient care. At this time, the IDEEL appears to be the most robust tool and should be considered for use in clinical practice and future research.”

- Please do not mention sentences that cannot be supported by data shown.

- First sentence of the conclusion is repeated information. Please delete it.

- Please see comments in the ABSTRACT.

6. PLOS authors have the option to publish the peer review history of their article (what does this mean?). If published, this will include your full peer review and any attached files.

Reviewer #1: No

Reviewer #2: No

---

## [Author Response · Author response to Decision Letter 0]

21 May 2021

Response to Reviewers:

PONE-D-20-39488

A systematic review assessing the quality of patient reported outcomes measures in dry eye diseases

Reviewer #1:

This study seeks to perform a formal search to identify patient-reported outcome measures in dry eye and ocular surface disease and to determine whether these instruments were developed using the COSMIN guidelines or what sort of reporting has been done in other studies for these various instruments. This is an important topic, as PROMs are really useful in dry eye and the evaluation of the quality of the instruments frequently used often seems lacking. Using the COSMIN standard to for evaluation here seems like a good idea.

We would like to thank the reviewer for their constructive comments.

R1-1 The abstract background states that the authors “evaluated the psychometric properties of studies” that considered PROMs, but I don’t think that’s quite right. The PROMs have psychometric properties, not the studies, right?

Thank you for your comment and sorry for the confusion. The COSMIN initiative described by Mokkink et al (2010) aimed to develop “a checklist containing standard for evaluating the methodological quality of studies on measurement properties”. Our aim was to apply this checklist and review the methodological quality of studies evaluating the psychometric properties of dry eye and ocular surface disease measures. It may not be clear from the paper that this was our approach. We have added more details about the COSMIN checklist in the method section where we have added more information about this. We hope this will clarify our approach. We have also reviewed the rest of the document to make sure this is clarified.

Please, see Ln 57-61:

“In general, psychometric properties such as content validity, measurement error and structural validity were not assessed by the studies included in this review. Studies reviewing The Impact of Dry Eye on Everyday Life (IDEEL) presented with the highest quality scores together with the Ocular Surface Disease Index (OSDI) questionnaire.”

Please, see Ln 65-69:

“The majority of the studies evaluating PROMs included in this review did not meet the recommended COSMIN criteria and the quality of the PROMs evaluated is not assured. Further evaluation of their psychometric properties is required if these are going to be used in clinical practice or research.”

Please, see Ln 171:

“Therefore, we present a systematic review of the quality of studies that report an evaluation of the psychometric properties of PROMs developed for use in patients with dry eyes and ocular surface diseases following the latest COSMIN guidelines (18).”

Please, see Ln 149-170 on where we have now included:

“Mokkink et al. (18) described COSMIN guidelines as a useful tool for the evaluation of the methodological quality of studies assessing the psychometric properties of PROMs. The guidelines were developed between 2006 and 2007 with input from a Delphi panel of more than 50 international experts who identified the key items and measurement properties captured by the guidelines. The guidelines were also updated in 2018 (19) when the authors decided to align with the Cochrane Collaboration for systematic reviews of intervention studies (20) by including inadequate studies which were previously excluded to avoid potential bias in their results. Additionally, the updated version removed reference to reasonable gold standard regarding criterion validity and responsiveness. The COSMIN panel reached a consensus that currently no gold standard exists for PROMs. Finally, the updated version removed references to sample size standards except for adequate sample size for Content Validity, Structural Validity and Cross-cultural validity. Using a comprehensive review such as the COSMIN guidelines can help determine which psychometric properties were considered important and the best way to evaluate these using a standardised approach. These guidelines are not only useful for research settings but clinicians can also benefit by using this practical tool to evaluate the potential of outcome measurement instruments in practice. Using these guidelines to evaluate the quality of the studies ensures a consistent approach for evaluating the quality of studies evaluating PROMs and allows structured comparisons to be carried out.”

R1-2 The authors tell us in 110-115 that a previous study by Grubbs on dry eye questionnaires identified the OSDI and IDEEL as the most reliable measure. They note that they think the omission of the term “ocular surface disease” in searching for instruments in that study was a flaw, though they come to similar conclusions in the end.

Thank you for your comment. “dry eye disease” is a feature of some but not all forms of “ocular surface disease” which is defined as a group of disorders, of diverse pathogenesis, in which disease results from the failure of mechanisms responsible for maintaining a healthy ocular surface. Ocular surface disease indicates damage to the surface layers of the eye. These include conditions with systemic comorbidities such as rheumatoid arthritis, Sjögren's syndrome, mucous membrane pemphigoid, Stevens-Johnson syndrome, atopic keratoconjunctivitis that can lead to ocular surface failure (conjunctivalisation, opacification and vascularisation of the cornea, and keratinisation of the cornea and conjunctiva followed by blindness). The spectrum also includes microbial keratitis (infections of the cornea) and ocular surface neoplasias. Given the broad-spectrum range of diseases, specific conditions are frequently excluded from ‘dry eye disease’ trials. While our methodological approach using the COSMIN guidelines confirmed Grubbs et al. findings, there are no specific instruments that capture the wellbeing of patients with complex ocular surface disease. We have strengthen the importance if this argument by inserting the following text: (Ln 133-146). 

“The term “ocular surface disease” was not included in the search strategy. “Ocular surface disease” is defined as a group of disorders, of diverse pathogenesis, in which disease results from the failure of mechanisms responsible for maintaining a healthy ocular surface. The term indicates damage to the surface layers of the eye and includes conditions with systemic comorbidities such as rheumatoid arthritis, Sjögren's syndrome, mucous membrane pemphigoid, Steven-Johnson syndrome, atopic keratoconjunctivitis that can lead to ocular surface failure and blindness. Other conditions include infective keratitis, and ocular surface tumours. While ‘Dry eye disease’ is a feature of some but not all forms of ocular surface disease, systemic comorbidities amongst those who have dry eye disease, can impact upon the patients’ tolerance of DED symptoms. The difficulty in delivering arduous treatment regimens in these patients impact upon patient wellbeing e.g. limited hand function can impinge upon their ability to self-administer eye drops and increases dependence on others. This may not be generalisable to other forms of ocular surface disease.

. 

R1-3 They also note that Grubbs et al. didn’t use the COSMIN guidelines, which they state is the gold standard for this type of work. I would note that the only reference supporting this gold standard statement is by the authors of the COSMIN guidelines. In general, some more discussion of how exactly the COSMIN guidelines are superior to other methods of evaluation would be useful.

Thank you we appreciate this is potentially biased and have removed reference to a gold standard and amended the introductory section (Ln 149-170) to demonstrate why we consider using these guidelines to be important.

“Mokkink et al. (18) described COSMIN guidelines as a useful tool for the evaluation of the methodological quality of studies assessing the psychometric properties of PROMs. The guidelines were developed between 2006 and 2007 with input from a Delphi panel of more than 50 international experts who identified the key items and measurement properties captured by the guidelines. The guidelines were also updated in 2018 (19) when the authors decided to align with the Cochrane Collaboration for systematic reviews of intervention studies (20) by including inadequate studies which were previously excluded to avoid potential bias in their results. Additionally, the updated version removed reference to reasonable gold standard regarding criterion validity and responsiveness. The COSMIN panel reached a consensus that currently no gold standard exists for PROMs. Finally, the updated version removed references to sample size standards except for adequate sample size for Content Validity, Structural Validity and Cross-cultural validity. Using a comprehensive review such as the COSMIN guidelines can help determine which psychometric properties were considered important and the best way to evaluate these using a standardised approach. These guidelines are not only useful for research settings but clinicians can also benefit by using this practical tool to evaluate the potential of outcome measurement instruments in practice. Using these guidelines to evaluate the quality of the studies ensures a consistent approach for evaluating the quality of studies evaluating PROMs and allows structured comparisons to be carried out.”

R1-4 I believe the authors could be considerably clearer about how they come to the conclusion that the IDEEL is the only instrument that meets the COSMIN standard. This is not an easy task, perhaps, considering that there are many COSMIN indicators and many different PROMs being considered. However, the reader needs to be able to understand what aspects of the methodological quality of studies on these instruments were judged inadequate. I don’t think the Results section really accomplishes this. For instance, my reading of the section on the Ocular Comfort Index does not turn up any obvious discussion of a problem. When I examine Table 3, again, nothing is particularly obvious.

Thank you for your comment. We have amended the section on Ln 552-555 and now it reads as follow: 

“Based on our assessments using the COSMIN checklist, the IDEEL study presented with the highest quality scores in terms of the evaluation of its psychometric properties (51). They also comply with current FDA guidelines (Supplementary Table 2) requiring PRO being used in labelling claims to ensure that they can demonstrate content validity with input from patients lived experiences of the health condition.”

Additionally, we have included more details on the results section of the ocular comfort index that now is amended on Ln 462-466: 

“Development of the PROM started with a literature review and patient interviews. However, it was not stated whether cognitive de-briefing interviews took place. Given the limited information we were unable to assess whether the content validity of the OCI met with the COSMIN criteria.”

Also, we have included in the discussion the following part that can be found on Ln 535-543:

“Many of the studies were also based on reviews using classical test theory and this is reflected in the COSMIN guidelines where common measurement standards (e.g. validity and reliability are evaluated using factor analysis and Cronbach’s Alpha). For example, while the content validity of the OCI could not be determined conclusively, it has good measurement properties and may be used for studies after initial cognitive work is done with patients. However, increasingly modern psychometric approaches such as IRT and Rasch analysis are being used to evaluate the extent to which interval level measurement and unidimensionality is being achieved (49). Many Rasch papers do not necessarily present CTT statistics as well as IRT or Rasch indices.

As more studies are using only Rasch based methods for evaluating PROMs it might be necessary to have additional guidance within the COSMIN guidelines to reflect this. This approach especially in the development of new PROMs can evaluate additional psychometric properties such as the evaluation of the scoring responses for disordered thresholds.”

R1-5 At line 177 the authors discuss the fact that an overall score is determined by taking the lowest rating of any box considered. Perhaps that score could be reported for each? Sticking with the OCI, in Table 3 I see N/A reported for internal consistency and a question mark for responsiveness (though in the Results text there is a responsiveness study described). Now, in the Structural Validity section of Table 3 for the three IDEEL studies, I see N/As, inadequates, or question marks. Was it the question mark rating that made that aspect acceptable? There must be a way to make the summary of the evaluation more easily accessible to the reader.

Thank you for your comment. To make it clearer we have included the previous Supplementary Table 1 in the main manuscript. This table is now table 4 and is displayed on Ln 256-257.

R1-6 Is the N/A for internal consistency the low score for the OCI? Those authors used Winsteps to conduct their Rasch analysis. They reported Rasch person reliability rather than Cronbach Alpha, as is typical in a Rasch study, but could’ve easily just reported the KR-20 value from the same Winsteps output table. If the authors of the OCI study had just reported Cronbach Alpha would that study have a good score for internal consistency? Actually, my reading of the COSMIN reference suggests that the Rasch stats presented in the OCI study should qualify as a measure of internal consistency: “For IRT-based scores: calculate standard error of theta (SE (θ)) or reliability coefficient of estimated latent trait value (index of (subject or item) separation) for each unidimensional scale or subscale”. So perhaps I am misunderstanding what N/A means in this table, but I would’ve rated the OCI study “very good” on internal consistency based on that quote from the COSMIN checklist reference. Could the authors clarify please?

Thank you for your comment. The authors have taken advice from a colleague with expertise in Winsteps and we would agree with your suggestion, although this is not covered by the COSMIN guidance as reflected in Table 4, further examination of the COSMIN checklist references would suggest this should be ‘very good’ and we have amended the text in the paper to reflect this. This information can be found on Ln 472-476 and it now it reads as:

“After the first Rasch analysis was performed and 3 misfitting items were removed, the reduced 12-item scale showed good fit. Fit statistics included INFIT MNSQ ranging from 0.86 to 1.17 and ZSTD from -2.0 to 2.4. OUTFIT MNSQ ranging from 0.75 to 1.20 and ZSTD from 0.61 to 0.73. The separation indices in the OCI demonstrated a good level of person/item separation with a person separation of 2.66 indicating stable item difficulty estimates and discrimination between 3 levels of ability across the scale (46) and an item separation of 11.12.”

Also, we do have amended Table 3 by changing NA with DNA as you can see now on Ln 250-253.

“Table 3 Results of studies on measurement properties”

“DNA = Data not available for assessment”

“PSI= Person separation index”

“ISI = item separation index”

Finally, we acknowledge the COSMIN guidelines limitation in considering the Rasch analysis for the quality of the PROM’s measurement properties and we have included this in the section Strength and Limitations on Ln 567-568 and it reads as follow:

“Also, COSMIN guidelines while are assessing the quality of the PROM’s measurement properties do not accommodate Rasch analysis and this should be seen as a limitation too.”

R1-7 Actually, the authors are quite fair about noting that many of the instruments evaluated were developed prior to the existence of COSMIN, and in many places are clear that the evaluation is of the studies on the instruments and not the instruments themselves. But it is very easy to slip into a discussion of which PROM is best, rather than which PROM has been evaluated in the best way. When the guidelines are specifically meant to evaluate the methodological quality of the STUDIES of these instruments, and the lowest score in any of the many evaluation categories is the determining factor, and the guidelines were developed after many of the instruments were, it is not clear to me that statements declaring the IDEEL as the only instrument that should be considered for use in clinic and research are particularly useful. Also, the actual content of the questions of the various surveys may or may not be suited for any individual study or clinical purpose.

We agree with your view. It is very easy to misjudge the aim of our study and slip into a discussion of which PROM is best. As you suggesting above, we have amended the sentence about the IDEEL on Ln 552-555 and now it reads as follow:

“Based on our assessments using the COSMIN checklist, the IDEEL study presented with the highest quality scores in terms of the evaluation of its psychometric properties (51). They also comply with current FDA guidelines (Supplementary Table 2) requiring PRO being used in labelling claims to ensure that they can demonstrate content validity with input from patients lived experiences of the health condition.”

R1-8 116: patients missing an apostrophe

Thank you. We have now amended this section. This change can be read on Ln 133-146 as:

“The term “ocular surface disease” was not included in the search strategy. “Ocular surface disease” is defined as a group of disorders, of diverse pathogenesis, in which disease results from the failure of mechanisms responsible for maintaining a healthy ocular surface. The term indicates damage to the surface layers of the eye and includes conditions with systemic comorbidities such as rheumatoid arthritis, Sjögren's syndrome, mucous membrane pemphigoid, Steven-Johnson syndrome, atopic keratoconjunctivitis that can lead to ocular surface failure and blindness. Other conditions include infective keratitis, and ocular surface tumours. While ‘Dry eye disease’ is a feature of some but not all forms of ocular surface disease, systemic comorbidities amongst those who have dry eye disease, can impact upon the patients’ tolerance of DED symptoms. The difficulty in delivering arduous treatment regimens in these patients impact upon patient wellbeing e.g. limited hand function can impinge upon their ability to self-administer eye drops and increases dependence on others. This may not be generalisable to other forms of ocular surface disease.”

R1-9 How many times were discrepancies between reviewers noted and how exactly were they resolved?

Thank you. There were no discrepancies between the reviewers noted but, as mentioned in Ln 208-209:

“At any stage, if the reviewers were unable to reach a consensus, a fourth reviewer was consulted (AS or SR).”

R1-10 184-5: What does this sentence mean, exactly? Perhaps it would be useful to note which important measurement properties have not been evaluated for each instrument?

Thank you for your comment. We have amended this section to include the important measurement properties that were not evaluated in the included studies. Now it reads on Ln 232-236 as follow: 

“This section presents information on the measurement properties that were reported by the individual studies including properties such as content validity, cross cultural validity, measurement invariance, measurement error and criterion validity.

R1-11 When the 127 records were assessed for eligibility, most were excluded as being out of scope. But there was a scope question in the previous step, in which thousands of records were eliminated. Were there differences in the evaluation of scope at these two stages? I would imagine that the 127 records, having made it through the previous round of eliminations, must’ve been fairly closely related to the target scope.

We thank the reviewer of the comment. There was no revision of scope at any time during the study. The initial exclusion was based on our assessment of the titles and abstracts. Although abstracts generally provide enough information to make decisions about inclusions and exclusions, sometimes a final decision can only be made after reading the full text of articles. So as not to exclude any potentially relevant articles, those that do not provide enough information in their abstracts are taken forward for full text screening. Upon reading the full texts of the 127 articles, we found that 113 did not actually meet our predefined criteria and so were excluded.

R1-12 There is some discussion of Rasch analysis in the paper, but no coverage of the differences between the Classical Test Theory and Item Response Theory approaches and how those differences are relevant to the entire COSMIN approach. I believe this would be beneficial for readers.

Thank you for your comment, we agree that a comparison would be useful but the nature of the paper and the extent of the discussion to do this justice might be prohibitive. However, we have added a brief comment (Ln 535-543) and added a reference that readers might want to read if they would like to have more information. We hope this addresses your comment.

“Many of the studies were also based on reviews using classical test theory and this is reflected in the COSMIN guidelines where common measurement standards (e.g. validity and reliability are evaluated using factor analysis and Cronbach’s Alpha). For example, while the content validity of the OCI could not be determined conclusively, it has good measurement properties and may be used for studies after initial cognitive work is done with patients. However, increasingly modern psychometric approaches such as IRT and Rasch analysis are being used to evaluate the extent to which interval level measurement and unidimensionality is being achieved (49). Many Rasch papers do not necessarily present CTT statistics as well as IRT or Rasch indices.

As more studies are using only Rasch based methods for evaluating PROMs it might be necessary to have additional guidance within the COSMIN guidelines to reflect this. This approach especially in the development of new PROMs can evaluate additional psychometric properties such as the evaluation of the scoring responses for disordered thresholds.” 

Reviewer #2: 

GENERAL CONSIDERATIONS

The authors have proposed to comprehensively evaluate the psychometric properties of studies that considered the use of PROMs in DED and OSD.

- Data is very interesting, however, some concepts and structural changes must be considered

Thank you for the positive comments.

R2-1- There rationale of the question of this systematic review must be clear for the readers. The purpose of the abstract seems to be different from the one at the end of the INTRODUCTION.

Please see below.

ABSTRACT

R2-2- The authors have mentioned: “comprehensively evaluated the psychometric properties of studies that considered the use of PROMs in DED and OSD.” The purpose of the abstract seems to be different than the one mentioned at the end of the INTRODUCTION. Please rewrite the purpose of the abstract according to the real purpose of the article.

We have amended the purpose of the abstract to match the real purpose of the article. Thank you. Now it reads on Ln 37-44 as follow: 

“Background

Patient-reported outcome measures (PROMs) can provide valuable insights on the impact of a disease or treatment on a patient’s health-related quality of life. In ophthalmology, particularly in dry eye disease (DED) and ocular surface disease (OSD), it is unclear whether the available PROMs were developed using comprehensive guidelines. To address this, we evaluated the methodological quality of studies assessing the psychometric properties of PROMs in DED and OSD [PROSPERO registration number CRD42019142328].

R2-3- The authors have mentioned as conclusion: “We suggest considering the IDEEL questionnaire, which showed to be the most robust tool in DED and OSD” Please answer the question proposed in the purpose. Suggesting that the questionnaire is robust was never an issue here. If it is, please change the rationale, sentence in the ABSTRACT and introduction. I would strongly suggest deleting this sentence or citing it in the discussion.

We have removed the sentence in the abstract and amended on Ln 64-65 as follow:

“The quality of studies evaluating PROMs in DED and OSD was considered using the COSMIN standards.”

R2-4- The authors have mentioned as conclusion: “The majority of the PROMs included in this review did not meet the proposed criteria and further validation work is required.” This answers the purpose of the study mentioned at the end of the INTRODUCTION.

Thank you for your comments.

R2-5- The authors have mentioned as conclusion: “PROMs in DED and OSD patients could offer a unique perspective and provide valuable insights in research and clinical settings to improve DED patient care.” Please understand that this is not a conclusion so I strongly suggest deleting it or citing it in the DISCUSSION.

Thank you. As suggested the sentence on Ln 69-70 was deleted. The sentence deleted was:

“PROMs in DED and OSD patients could offer a unique perspective and provide valuable insights in research and clinical settings to improve DED patient care.” 

DISCUSSION

R2-6- The authors have mentioned: “we present a systematic review of studies that report an evaluation of psychometric properties of PROMs developed for use in patients with dry eyes and ocular surface diseases following the latest COSMIN guidelines [18].”

Thank you.

R2-7- The authors have mentioned as conclusion: “The quality of different studies evaluating PROMs being used to evaluate the impact of DED and its treatment in patients QoL were reviewed against the exacting COSMIN standards. The majority of those included in this review did not meet the proposed criteria and further validation work is required. PROMs in DED and OSD patients could offer a unique perspective and provide valuable insights in research and clinical settings to improve DED and OSD patient care. At this time, the IDEEL appears to be the most robust tool and should be considered for use in clinical practice and future research.”

Thank you for your comment. We have amended the last sentence and it now reads on Ln 545-548 as: 

“Based on our assessments using the COSMIN checklist, the IDEEL study presented with the highest quality scores in terms of the evaluation of its psychometric properties (51). They also comply with current FDA guidelines (Supplementary Table 2) requiring PRO being used in labelling claims to ensure that they can demonstrate content validity with input from patients lived experiences of the health condition.”

R2-7- Please do not mention sentences that cannot be supported by data shown.

Thank you. See above comment and its amendment.

R2-8- First sentence of the conclusion is repeated information. Please delete it.

The first sentence of the conclusion has been deleted. Thank you for your comment.

R2-9- Please see comments in the ABSTRACT.

Thank you. All the comments have been addressed.

---

## [Decision Letter · Decision Letter 1]

15 Jun 2021

A systematic review assessing the quality of patient reported outcomes measures in dry eye diseases

PONE-D-20-39488R1

Dear Dr. Rauz,

We’re pleased to inform you that your manuscript has been judged scientifically suitable for publication and will be formally accepted for publication once it meets all outstanding technical requirements.

Kind regards,

Peter M. ten Klooster, Ph.D.

Academic Editor

PLOS ONE

Additional Editor Comments (optional):

Both reviewers' comments have been adequately adressed. Most importantly, the actual objective of the study is now more clearly formulated throughout the manuscript and matches with the actual conlusions drawn.

Reviewers' comments:

Reviewer's Responses to Questions

**Comments to the Author**

1. If the authors have adequately addressed your comments raised in a previous round of review and you feel that this manuscript is now acceptable for publication, you may indicate that here to bypass the “Comments to the Author” section, enter your conflict of interest statement in the “Confidential to Editor” section, and submit your "Accept" recommendation.

Reviewer #1: All comments have been addressed

2. Is the manuscript technically sound, and do the data support the conclusions?

Reviewer #1: Yes

3. Has the statistical analysis been performed appropriately and rigorously? 

Reviewer #1: Yes

4. Have the authors made all data underlying the findings in their manuscript fully available?

Reviewer #1: Yes

5. Is the manuscript presented in an intelligible fashion and written in standard English?

Reviewer #1: Yes

6. Review Comments to the Author

Reviewer #1: (No Response)

7. PLOS authors have the option to publish the peer review history of their article (what does this mean?). If published, this will include your full peer review and any attached files.

Reviewer #1: No

---

## [Editor Report · Acceptance letter]

29 Jul 2021

PONE-D-20-39488R1 

A systematic review assessing the quality of patient reported outcomes measures in dry eye diseases 

Dear Dr. Rauz:

I'm pleased to inform you that your manuscript has been deemed suitable for publication in PLOS ONE. Congratulations! Your manuscript is now with our production department. 

Kind regards, 

on behalf of

Dr. Peter M. ten Klooster 

Academic Editor

PLOS ONE